# Which elements of hospital-based clinical decision support tools for the assessment and management of children with head injury can be adapted for use by paramedics in prehospital care? A systematic mapping review and narrative synthesis

Alyesha Proctor ,[1] Mark Lyttle,[2] Jedd Billing,[1] Pauline Shaw,[1] Julian Simpson,[1] Sarah Voss ,[3] Jonathan Richard Benger[4,5]

For numbered affiliations see end of article.

**Correspondence to**
Alyesha Proctor;
alyesha.proctor@uwe.ac.uk

## ABSTRACT

**Objective** Hospital-based clinical decision tools support clinician decision-making when a child presents to the emergency department with a head injury, particularly regarding CT scanning. However, there is no decision tool to support prehospital clinicians in deciding which head-injured children can safely remain at scene. This study aims to identify clinical decision tools, or constituent elements, which may be adapted for use in prehospital care.

**Design** Systematic mapping review and narrative synthesis.

**Data sources** Searches were conducted using MEDLINE, EMBASE, PsycINFO, CINAHL and AMED.

**Eligibility criteria** Quantitative, qualitative, mixed-methods or systematic review research that included a clinical decision support tool for assessing and managing children with head injury.

**Data extraction and synthesis** We systematically identified all in-hospital clinical decision support tools and extracted from these the clinical criteria used in decision-making. We complemented this with a narrative synthesis.

**Results** Following de-duplication, 887 articles were identified. After screening titles and abstracts, 710 articles were excluded, leaving 177 full-text articles. Of these, 95 were excluded, yielding 82 studies. A further 14 studies were identified in the literature after cross-checking, totalling 96 analysed studies. 25 relevant in-hospital clinical decision tools were identified, encompassing 67 different clinical criteria, which were grouped into 18 categories.

**Conclusion** Factors that should be considered for use in a clinical decision tool designed to support paramedics in the assessment and management of children with head injury are: signs of skull fracture; a large, boggy or non-frontal scalp haematoma neurological deficit; Glasgow Coma Score less than 15; prolonged or worsening headache; prolonged loss of consciousness; post-traumatic seizure; amnesia in older children; non-accidental injury; drug or alcohol use; and less than 1 year old. Clinical criteria that require further investigation include mechanism of injury, clotting impairment/anticoagulation, vertigo, length of time of unconsciousness and number of vomits.

## STRENGTHS AND LIMITATIONS OF THIS STUDY

⇒ The review is highly inclusive, with a range of global study settings, including qualitative, quantitative and mixed-methods research.

⇒ The review provided an opportunity to complete a detailed and contemporary search of all available sources and identified 25 in-hospital tools, compared with 14 in a previous systematic review.

⇒ The role of mapping reviews is to provide a descriptive account of the published literature; therefore, this should be considered when assessing the findings of the overall evidence synthesis.

⇒ A direct comparison of the clinical decision support tools is not possible because they address different questions, age groups, injury severities and outcomes.

## INTRODUCTION

Healthcare systems internationally are under substantial pressure due to rising patient demand. A leading priority is to reduce the number of patients who are unnecessarily conveyed to an emergency department (ED) by ambulance.[1] Children frequently use the ambulance service and ED, and childhood head injury is common; approximately 800 000 head-injured children attend EDs each year in the UK.[2] Very few children with head injury who are conveyed to the ED by ambulance need specialist treatment,

and the vast majority could be managed safely at scene. Despite this, a national overview of head injury in children found that one-third of children in hospital with head injury were transported by ambulance, and 74% of those conveyed were non-serious, requiring no intervention.[3]

Head injury guidance produced by the National Institute for Health and Care Excellence (NICE)[4] is of limited utility to paramedics for reasons including: a lack of validation in prehospital care; reference to patient observation over time which is not always possible; challenges in the assessment of amnesia in younger children; and limited research evidence on which to base recommendations relating to ambulance non-conveyance.[4]

There are several existing validated hospital-based decision support tools that clinicians use when a child presents to the ED with a minor head injury, and studies have demonstrated their effectiveness in aiding decision-making for both immediate discharge and CT scanning.[5 6] Currently, there are no prehospital decision tools designed to support paramedics in reducing unnecessary conveyance to the ED for head-injured children. None of the existing hospital-based tools can be implemented directly in prehospital care, since they are designed to support decision-making on performing CT scanning, rather than ED conveyance.

### Aim

To systematically identify and examine clinical criteria within existing hospital-based clinical decision support tools for children with minor head injury, with a focus on determining their potential to aid decision-making regarding hospital conveyance by paramedics.

## METHODS
### Patient and public involvement

Patients and public have been involved throughout this research. Both a Young Persons Advisory Group (children aged 9–16 years) and Parent Advisory Group have supported this study and confirmed its importance for children, parents, and the wider health and care system.

### Introduction

Systematic mapping aims to collate, explain and catalogue available evidence (primary and secondary) relating to a topic of interest, to identify a focus for more specific investigations.[7] It is a review methodology that is often used in healthcare research that aims to detail and categorise literature on a particular topic, with the intention of developing this into more comprehensive work.[8] This systematic mapping review was conducted in accordance with published methods[9] and is reported in line with the Preferred Reporting Items for Systematic Reviews and Meta-Analyses (PRISMA) statement.[10] It has not been registered with PROSPERO as it is a mapping review.

### Search strategy

Searches were conducted in the following databases, for articles published between January 1980 and August 2022, to ensure that all relevant contemporary research was included: MEDLINE, EMBASE, PsycINFO, CINAHL and AMED (online supplemental material: search strategy). A Google Scholar and a Web of Science search were undertaken to identify reports or proceedings not indexed in those mentioned above. A literature advisory group, including experts in the field, was consulted to identify relevant grey literature and unpublished reports. Furthermore, to find additional published, unpublished and ongoing studies, the reference lists of all relevant studies were examined.

Search terms were developed iteratively by discussion among the research team, patient and public representatives and a librarian, seeking a balance between comprehensiveness and focus. A combination of Medical Subject Headings terms and synonym text-strings/phrases was used in the search strategy and these were combined using Boolean operators. Updated searches were rerun before the final analysis. The search terms used can be seen in table 1.

The search was limited to English-language studies due to resource restrictions and the cost of translation, and only articles with full texts available were included. Studies that may have been relevant but did not include or refer to a clinical decision support tool for the assessment and management of head-injured children were excluded; however, there were no restrictions on the types of studies included. Eligibility criteria are presented in full in table 2.

### Extracting, coding and synthesising the data

Titles and abstracts were double-screened for relevance and assessed for eligibility using Covidence software,[11] following which full texts of included articles underwent review for eligibility. Any disagreements were resolved through discussion with a senior researcher. Reasons for exclusion were documented, a topic expert reviewed the final set of included papers to ensure relevant studies and tools had not been overlooked. Figure 1 shows the PRISMA flow diagram.[10]

Data extraction included systematically identifying the title of each tool, the aim, derivation reference, supporting literature, conclusion, and key points relating to the content and format of a clinical decision support tool for paramedic use. An inductive coding frame was developed to map emerging concepts, which involved deriving codes from the data. Key messages/concepts in relation to predicting conveyance from all studies were extracted from the methods, results and conclusion sections. This resulted in a visual synthesis and classification of available studies and tools. The data extraction tool was piloted using two included studies, and data were extracted by hand and cross-checked by another researcher.

Clinical criteria used in decision-making were extracted from all tools; a summary map was produced providing

**Table 1**  Search terms

| Clinical decision support tool | Head injury | Children | Hospital |
|---|---|---|---|
| Clinical decision tool | Minor head injury | Child | ED |
| Clinical decision rule | TBI (Traumatic brain injury) | Pediatric | Emergency Department |
| Diagnostic accuracy tool | Head trauma | Paediatric | A+E |
| Triage tool | Head wound | Baby | Accident and Emergency |
| Hospital-based tool | Intracranial injury | Babies | Emergency room |
| CDRs | CiTBI (clinical important traumatic brain injury) | Infant | Trauma centre |
| Intervention | ICI (intracranial injury) | Schoolchild | |
| Clinical decision score | CSII (clinically significant intracranial injury) | Adolescent | |
| Paediatric head injury predictive tool | Trivial head injury | Teenager | |
| Clinical Prediction Rule | | Young person | |
| Clinical Prediction Tool | | | |
| Clinical decision support tool (CDST) | | | |

a visual display of clinical criteria included in each tool. After independently producing pilot categories based on a sample of clinical decision support tools, the research team formed a consensus on categories. The frequency of each clinical criterion was tabulated, and performance accuracy was recorded where reported. A narrative synthesis was subsequently developed using this framework.

### Assessment of quality

No formal assessment of quality was undertaken, as this is not required in mapping reviews.[12] Data extraction in mapping reviews describes the studies and their main methods but does not attempt a full data extraction and quality assessment.[12]

**Table 2**  Inclusion and exclusion criteria

| Inclusion criteria | Exclusion criteria |
|---|---|
| Studies in the English language | Non-English language |
| Quantitative, qualitative, mixed-methods research, systematic reviews | |
| Children (under 19 years of age) | Adult population (19 years or older) |
| Date of publication 1980–present | |
| Any study that includes a clinical decision support tool for assessing and managing children with head injury | Any study that did not include a clinical decision support tool for the assessment and management of head-injured children |
| International as long as in the English language | |

### RESULTS

A total of 887 articles were identified after duplicates were removed. After screening titles and abstracts, 710 articles were excluded, which left 177 full-text articles to be assessed for eligibility. Of these, 95 were excluded for reasons including: adult population only, not including a clinical decision support tool, not in the English language and full text not available. Three articles were not included, as they were conference abstracts. Fourteen additional studies were identified through citation searching, resulting in 96 included studies (figure 1).

In total, 25 relevant in-hospital clinical decision support tools for the assessment and management of children with head injury were identified, which can be seen in table 3.

Sixty-seven decision-making criteria were extracted across all tools. Several criteria related to the same clinical feature (for example, vomiting); however, there were many different cut-off points (eg, vomiting once only, vomiting twice or vomiting three times). These 67 criteria were grouped into 18 clinical categories: vomiting; headache; loss of consciousness (LOC); seizure; amnesia; dizziness/vertigo; clotting impairment/anticoagulated; non-accidental injury; age; severe mechanism of injury; drug/alcohol use; signs of a skull fracture; haematoma; neurological deficit; altered mental status; Glasgow Coma Score (GCS); drowsiness and well with no clinical concern.

Online supplemental table 1 shows which clinical categories are included within each hospital-based tool.

Of these clinical categories, the most frequent were skull fracture, LOC and scalp haematoma. The most infrequently used included vertigo, suspicion of non-accidental injury and patient acting normal as per parent. Table 4 tabulates the frequency of each clinical criterion.

The mix of clinical criteria used and their outcomes were similar but not identical for any of the tools. In

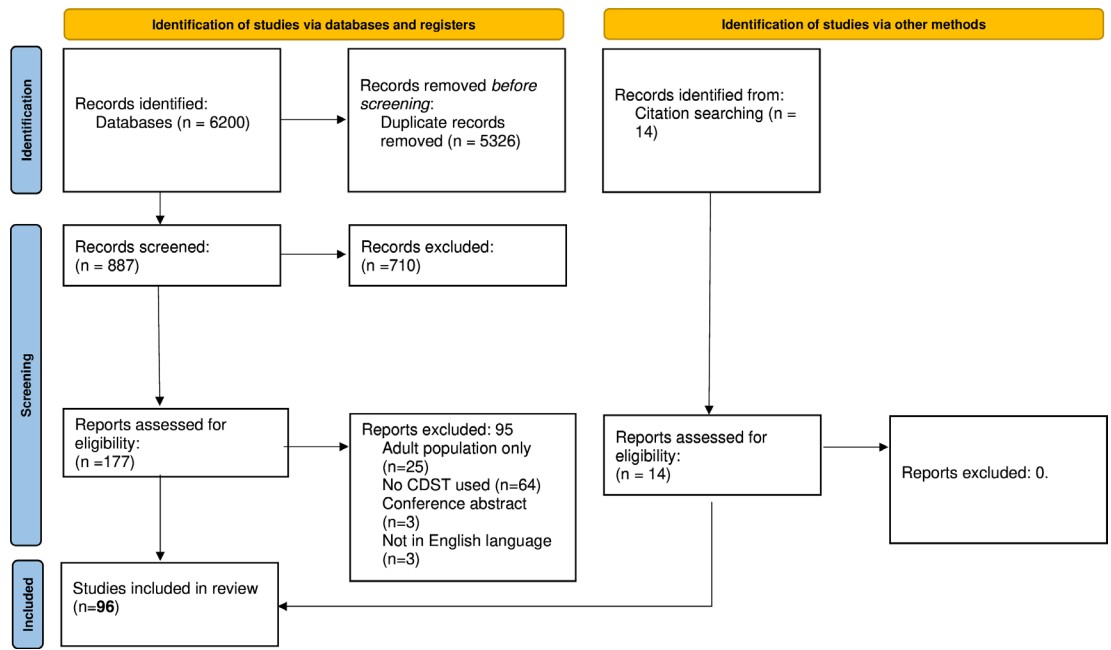

**Figure 1** Preferred Reporting Items for Systematic Reviews and Meta-Analyses flow diagram.[60] CDST, clinical decision support tool.

addition, the derivation studies used different populations and sample sizes; therefore, the performance accuracy of the tools relating to the potential to correctly rule out clinically significant brain injury in head-injured children was variable (online supplemental table 2).

Most tools displayed high sensitivities (>90%), while specificities ranged from 15% to 87%. PECARN (Paediatric Emergency Care Applied Research Network rule) was the most widely externally validated, with positive conclusions drawn from research examining post-implementation of the tool.[13] However, PECARN results in a considerable number of normal scans being performed to identify one intracranial injury.[14] The Head Injury Discharge At Triage questionnaire (HIDATq) was designed for nurses to use in triage to safely discharge children with head injury before admission, which more closely resembles the aim of a prehospital tool designed to inform which children do not need to be conveyed to hospital. A narrative synthesis is presented below.

### Signs of skull fracture

This was the most frequently used clinical decision-making criterion (21 of 25) (84%). Variability of terms included: open or depressed skull fracture, basal skull fracture, tense fontanelle, palpable and unclear skull fracture. The only tools that do not include signs of skull fracture are HAYDEL (New Orleans/Charity Head Trauma/Injury Rule), DIET-RICH, PredAHT (Predicting Abusive Head Trauma) and Novel Simplified CDR (Clinical Decision Rule). All three of PECARN, CHALICE (Children's Head injury Algorithm for the prediction of Important Clinical Events Rule) and CATCH (Canadian Assessment of Tomography for Childhood Head injury Rule)

(the three most widely validated tools) included signs of skull fracture. PredAHT does not include signs of skull fracture despite the aim of this tool being to identify abusive head trauma. In contrast, Pedi-BIRN (Paediatric Brain Injury Research Network) which aims to identify children with abusive head trauma does include signs of skull fracture as a clinical predictor.[15] The Kids Intracranial Injury Decision Support tool for Traumatic Brain Injury concluded that a depressed skull fracture was a major risk factor for significant brain injury.[16] Babl et al[17] and Da Dalt et al[18] found that a basal, open or depressed skull fracture was most likely to trigger a CT response out of any other clinical criteria. Similarly, Quayle et al[19] concluded that the relative risk of intracranial injury is increased almost fourfold in the presence of a skull fracture, although the absence of skull fracture does not rule out intracranial injury. The CIDSS2 (Coma scale; Intoxication; Deficit (neurological); Seizure; Skull fracture 2) score,[20] a tool to help physicians to identify the minority of children with minor traumatic brain injury at increased risk of extended inpatient stay uses a scoring system; all clinical criteria score 1, apart from signs of skull fracture which score 2, suggesting it is deemed the most significant clinical predictor of significant brain injury. Many of the tools, such as NEXUS 2 (National Emergency X-Ray Utilization Study), describe signs and symptoms of a skull fracture, rather than explicitly stating 'signs of a skull fracture', for example, periorbital or periauricular ecchymoses, haemotympanum, and clear fluid from ears or nose.[21]

**Table 3** Hospital-based clinical decision support tools for the assessment and management of children with head injury

| CDST | Full title | Reference |
|---|---|---|
| PECARN | Paediatric Emergency Care Applied Research Network rule | Kuppermann et al[6] |
| CHALICE | Children's Head injury Algorithm for the prediction of Important Clinical Events Rule | Dunning et al[24] |
| CATCH | Canadian Assessment of Tomography for Childhood Head injury Rule | Osmond et al[29] |
| NEXUS 2 | National Emergency X-Ray Utilization Study | Oman et al[61] |
| HIDATq | Head Injury Discharge At Triage questionnaire | Aldridge et al[5] |
| PALCHAK | UC Davis Rule for Paediatric Head Trauma | Palchak et al[23] |
| HAYDEL | New Orleans/Charity Head Trauma/Injury Rule | Haydel and Shembekar[62] |
| ATABAKI | No title | Atabaki et al[63] |
| GREENES | No title | Greenes and Schutzman[22] |
| KLEMETTI | No title | Klemetti et al[27] |
| QUAYLE | No title | Quayle et al[19] |
| DIETRICH | No title | Dietrich et al[64] |
| GUZEL | No title | Güzel et al[65] |
| PredAHT | Predicting Abusive Head Trauma | Cowley et al[49] |
| CHIDA | Children's Intracranial Injury Decision Aid | Neumayer et al[66] |
| CIDSS2 | Stands for: Coma scale Intoxication Deficit (neurological) Seizure Skull fracture 2 | Greenberg et al[20] |
| PediBIRN | Paediatric Brain Injury Research Network | Pfeiffer et al[15] |
| Head CT Choice | Head CT Choice | Hess et al[67] |
| SNC Tool | The Scandinavian Neurotrauma Committee Guidance | Unden et al[25] |
| BIG-1 | Brain Injury Guidelines | Schwartz et al[68] |
| Novel Simplified CDR | Novel Simplified Clinical Decision Rule | Yogo et al[31] |
| Head Trauma EBG Algorithm | Head Trauma EBG Algorithm | Stopa et al[69] |
| DA DALT | No title | Da Dalt et al[18] |
| BUCHANICH | No title | Buchanich[70] |
| KIIDS-TBI-CDS | The Kids Intracranial Injury Decision Support tool for Traumatic Brain Injury | Greenberg et al[16] |

CDST, clinical decision support tool.

## Scalp haematoma

Scalp haematoma is used in 17 of the tools identified. However, many specify 'non-frontal' haematomas only, such as the NEXUS 2 tool.[21] Others (GREENES) support inclusion of non-frontal haematomas only; the authors found that non-frontal scalp haematomas had the strongest association with presence of intracranial haemorrhage.[22] The authors of PALCHAK (UC Davis Rule for Paediatric Head Trauma) concluded a scalp haematoma is more significant in children under 2 years.[23] The HIDATq tool does not stipulate location of haematoma; however, the authors do specify a size (over 5 cm).[5] Similarly, CHALICE only specifies size, which is also over 5 cm, but this is for children under 1 year old only.[24] PECARN clearly specifies 'temporal, occipital or parietal haematoma', and this is for the children under 2 years old.[6] The CATCH tool includes 'large boggy scalp haematoma'; however, it is unclear what constitutes 'large' as it does not specify what size or the location. Most excluded haematomas to the face.

## Loss of consciousness

LOC has different cut-off points across the identified clinical decision tools, including 'any LOC', 'LOC for over 5 s' and 'LOC for over 5 min'. It is the second most frequently used clinical predictor, appearing in 17 of the 25 clinical decision tools. The authors of PALCHAK[23] found that the risk of a traumatic brain injury was higher in those

**Table 4** Frequency of each clinical category across all the tools

| Categories | CDST | Frequency (n=25) |
|---|---|---|
| Signs of any description of skull fracture (depressed, basal skull, palpable) | PECARN 18, PECARN 2, NEXUS 2, HIDATq, PALCHAK, CHALICE, CATCH, GUZEL, BUCHANICH, DA DALT, KLEMETTI, QUAYLE, ATABAKI, SNC Guideline, BIG-1, Head trauma EBG Algorithm, GREENES, CHIDA, CIDSS2, PediBIRN, KIIDS-TBI-CDS | 21 (84%) |
| LOC | PECARN 18, PECARN 2, CHALICE, CATCH, NEXUS 2, ATABAKI, KLEMETTI, QUAYLE, HIDATq, GUZEL, DIETRICH, DA DALT, BUCHANICH, SNC Guideline, BIG-1, Novel Simplified CDR, Head Trauma EBG Algorithm | 17 (68%) |
| Scalp haematoma/trauma | PECARN 2, CHALICE, PALCHAK, KLEMETTI, NEXUS 2, GUZEL, HAYDEL, GREENES, CATCH, HIDATq, BUCHANICH, ATABAKI, SNC Guideline, Head Trauma EBG Algorithm, PredAHT, PediBIRN, KIIDS-TBI-CDS | 17 (68%) |
| Neurological deficit | CHALICE, NEXUS 2, HIDATq, GUZEL, DIETRICH, DA DALT, BUCHANICH, KLEMETTI, ATABAKI, QUAYLE, Head CT Choice, SNC Guideline, BIG-1, Head Trauma EBG Algorithm, CIDSS2 | 15 (60%) |
| Behaviour change/altered mental status | PECARN 18, PECARN 2, CATCH, HIDATq, PALCHAK, ATABAKI, BUCHANICH, DA DALT, KLEMETTI, QUAYLE, NEXUS 2, Head CT Choice, SNC Guideline, Novel Simplified CDR, Head Trauma EBG Algorithm | 15 (60%) |
| GCS | PECARN 18, PECARN 2, CHALICE, NEXUS 2, CATCH, HIDATq, ATABAKI, CHIDA, CIDSS2, Head CT Choice, SNC Guideline, Novel Simplified CDR, Head Trauma EBG Algorithm, DA DALT, KIIDS-TBI-CDS | 15 (60%) |
| Vomiting | CHALICE, NEXUS 2, PECARN 18, PALCHAK, HIDATq, HAYDEL, DIETRICH, BUCHANICH, DA DALT, Head CT Choice, SNC Guideline, Head Trauma EBG Algorithm | 12 (48%) |
| Headache | PECARN 18, CATCH, PALCHAK, GUZEL, DIETRICH, HAYDEL, ATABAKI, BUCHANICH, DA DALT, Head CT Choice, SNC Guideline, Novel Simplified CDR | 12 (48%) |
| Seizure | CHALICE, HIDATq, HAYDEL, ATABAKI, QUAYLE, DIETRICH, GUZEL, PredAHT, CIDSS2, SNC Guideline, Novel Simplified CDR, DA DALT | 12 (48%) |
| Severe mechanism of injury | PECARN 18, PECARN 2, GUZEL CHALICE, CATCH, HIDATq, ATABAKI, Novel Simplified CDR, Head Trauma EBG Algorithm | 9 (36%) |
| Drowsiness | CHALICE, HIDATq, DA DALT, PECARN 18, PECARN 2, NEXUS 2, ATABAKI, Head CT Choice | 8 (32%) |
| Amnesia | CHALICE, CATCH, NEXUS 2, HIDATq, HAYDEL, ATABAKI, GUZEL, DA DALT | 8 (32%) |
| Drug or alcohol use | HIDATq, HAYDEL, ATABAKI, CIDSS2, BIG-1 | 5 (20%) |
| Age | PECARN, HIDATq, ATABAKI, HAYDEL | 4 (16%) |
| Clotting impairment/anticoagulated | NEXUS 2, HIDATq, SNC Guideline | 3 (12%) |
| Dizziness/vertigo | ATABAKI, KLEMETTI | 2 (8%) |
| Non-accidental injury | CHALICE, HIDATq | 2 (8%) |
| Patient alert and well with no clinical concern/acting normal as per parent | PECARN 2, HIDATq | 2 (8%) |

BIG-1, Brain Injury Guidelines; CATCH, Canadian Assessment of Tomography for Childhood Head injury Rule; CDST, clinical decision support tool; CHALICE, Children's Head injury Algorithm for the prediction of Important Clinical Events Rule; CHIDA, Children's Intracranial Injury Decision Aid; CIDSS2, Coma scale; Intoxication; Deficit (neurological); Seizure; Skull fracture 2; GCS, Glasgow Coma Score; HAYDEL, New Orleans/Charity Head Trauma/Injury Rule; HIDATq, Head Injury Discharge At Triage questionnaire; KIIDS-TBI-CDS, Kids Intracranial Injury Decision Support tool for Traumatic Brain Injury; LOC, loss of consciousness; NEXUS 2, National Emergency X-Ray Utilization Study; Novel Simplified CDR, Novel Simplified Clinical Decision Rule; PALCHAK, UC Davis Rule for Paediatric Head Trauma; PECARN, Paediatric Emergency Care Applied Research Network rule; PediBIRN, Paediatric Brain Injury Research Network; PredAHT, Predicting Abusive Head Trauma; SNC Tool, Scandinavian Neurotrauma Committee Guidance.

with a history of LOC. However, in those children with an isolated LOC, there were no traumatic brain injuries. This is recognised in Kuppermann *et al*'s derivation study for PECARN[6]; the risk of clinically important traumatic brain injury was substantially lower than 1% in children with isolated LOC. PECARN includes LOC for over 5 s in the cohort under 2 years of age and LOC of unspecified duration for the cohort over 2 years. Differing to PECARN,

CHALICE uses the duration of over 5 min of LOC in any child under 16 years of age, and HIDATq does not specify the duration of LOC.

## Vomiting

This clinical criterion is a predictor in 12 of 25 tools, with four varying descriptors including 'any post-injury vomiting', 'over one vomit', 'over two vomits' or 'over three vomits'. PECARN includes 'any vomiting' for the cohort over 2 years old, as does the HIDATq tool, whereas CHALICE takes a different, less risk averse approach including 'three or more'.[24] Both CHALICE and PECARN have a high sensitivity and acceptable specificity for recognising clinically important traumatic brain injury, suggesting that both tools are appropriate for assessing mild head injury in children in the ED, despite the varying descriptions of vomiting. NEXUS 2 includes 'over one vomit' and CATCH stipulates 'over two vomits', but as eligibility criterion.

## Neurological deficit

For the purposes of this narrative description, neurological deficit will also include focal neurology, GCS, altered mental status, behavioural change, drowsiness and vertigo.

The research evidence that underpinned the DA DALT tool concluded that neurological deficit is one of the predictors that significantly increases the likelihood of intracranial injury.[18] Similarly, the CIDSS2 score concluded that neurological deficit was among the most important predictors for identifying whether paediatric patients should be admitted to hospital with a head injury.[20] There are varying descriptions to define a neurological deficit across the 25 tools, and there is some crossover with altered mental status/behavioural change, drowsiness and GCS. Some terms include vision changes, gait changes, repetitive questioning/speech, delayed response to external stimuli, sensory deficit, disorientation or just stating 'neurological deficit'. Authors of the Scandinavian Neurotrauma Committee (SNC) guidelines suggest that the clinical criterion 'altered mental status' is too complicated to use effectively, with potential to lead to unacceptable increases in CT rates.[25] Both PECARN (under 2) and HIDATq include 'patient well with no clinical concern/acting normal as per parent', which suggests no neurological deficit.

Many tools opt to use 'abnormal mental status' with varying descriptors, such as confusion, somnolence, repetitive or slow to respond to verbal communication. A GCS under 15 is associated with the presence of intracranial acute pathologies in young children with minor blunt head trauma.[26] GCS is a clinical criterion in 15 of the 25 identified tools, including PECARN, CHALICE and CATCH.

KLEMETTI is one of the only tools to include 'vertigo' as a clinical criterion; the authors found this variable was among one of the most significant to predict severely complicated head trauma in their multivariate logistic regression analysis.[27] ATABAKI is the only other tool to include vertigo as a clinical criterion; however, both these tools have low specificity and were not evaluated in a published validation study after development.

## Headache

Headache is a clinical predictor in only half of the 25 in-hospital tools. This may be because it is challenging to assess for a headache in younger children; they may express pain differently with signs such as irritability and persistent crying.[28] The authors of the BUCHANICH tool use headache as a clinical predictor, despite an inclusion criterion of age under 3 years only; however, this tool is not validated. PECARN includes headache for the cohort aged 2–18 years, but not for those aged under 2 years. For children who can verbalise or show that they have a headache, it is important to consider severity and timing of the headache. Some tools specify 'severe headache', 'prolonged headache' or 'worsening headache'. Worsening headache is associated with intracranial injury, being deemed a 'high risk' criterion in CATCH.[29] The HIDATq tool does not include headache as a clinical predictor, despite presenting the tool as 'ultra-safe', although it does include 'irritability' which could indicate pain in a younger child.

## Seizure

Just under half of the 25 in-hospital tools included seizure as a clinical predictor despite a post-traumatic seizure being a high determinant for an intracranial injury.[30] The CIDSS2 score found seizure to be one of the most significant criteria to identify whether patients should be admitted with head injury. PECARN is the most widely validated and implemented tool; however, it does not include seizure as a specific clinical predictor, compared with CHALICE and HIDATq, which do.

## Severe mechanism of injury

Nine of the 25 identified clinical decision support tools include 'severe or dangerous mechanism of injury' as a clinical predictor. The description of what constitutes a 'severe' mechanism of injury varies across the tools, from different levels of fall from height, road traffic collisions, to specific considerations such as a bicyclist not wearing a helmet who is struck by a car; however, fall from height is the most common mechanism of injury in children. The SNC guideline[25] does not include high-energy trauma mechanism as a strict risk factor, as these patients are relatively uncommon in Scandinavia and are managed according to separate clinical trauma protocols. The authors and creators of the SNC guidelines judged 'mechanism of injury' as a complicated clinical predictor to use, due to having a specific definition and often including assessment of fall height, vehicle speed and number of stairs.[25] Similarly, the Novel Simplified Tool[31] suggests that mechanism of injury is not considered reliable, and that it may be an overestimation to emphasise the injury mechanism as a predictor. The aim of this

clinical decision tool is to provide a simplified version of PECARN, CATCH and CHALICE (all of which include varying descriptors for mechanism of injury), so that it can be easily and quickly used for paediatric patients in busy emergency settings. Therefore, one distinguishing feature is that that this tool includes only one predictor for mechanism of injury, however; the Novel CDR misclassified eight patients as low risk.

### Amnesia

The authors of PALCHAK conclude that elimination of amnesia as a clinical predictor may decrease unnecessary CT scanning.[23] However, amnesia is a clinical predictor in 8 of the 25 hospital-based tools identified within this review. It is not possible to assess amnesia in non-verbal children and it is difficult to assess in young children[18] which is perhaps why PECARN does not include amnesia as a clinical predictor; however, all of CHALICE, CATCH, NEXUS 2 and HIDATq do. One study adapted the PECARN tool by introducing amnesia alongside other clinical predictors.[13] The authors concluded that the adapted PECARN rule was successfully implemented in an Italian paediatric ED; its use determined a low CT scan rate that was unchanged compared with previous clinical practice and showed an optimal safety and high efficacy profile.

### Non-accidental injury

Non-accidental injury is only included as a clinical predictor in two of the in-hospital clinical decision support tools (HIDATq and CHALICE), although some tools include non-accidental injury as an exclusion criterion, such as the Head Trauma EBG Algorithm and CATCH. Despite this, non-accidental injury is one of the biggest predictors of brain injury, particularly in children under the age of 2 years.[32] PECARN is not especially sensitive for identifying physically abused children.[33] There is no gold-standard diagnostic test to identify abusive head trauma though PredAHT and PediBIRN are specifically designed tools that can help clinicians to recognise cases of abusive head trauma.[33]

### Clotting impairment

Only 3 of the 25 in-hospital tools include clotting/impairment as a clinical predictor (HIDATq, the SNC guideline and NEXUS 2), since it is uncommon for a child to be on these types of drugs or have bleeding diatheses. The HIDATq tool is derived using the NICE head injury guidance and is designed to be 'ultra-safe' since the aim of the tool is to discharge children with head injury at triage, and therefore includes 'anticoagulant use/clotting impairment'. None of the three most widely validated tools (PECARN, CATCH and CHALICE) include 'anticoagulant use/clotting disorder' as a clinical predictor, because it was too rare to be of use.

### Drug and alcohol use

It is well-known that alcohol increases the risk of suffering a head injury due to its effects on balance and coordination.[34] Children with suspected drug or alcohol intoxication may be more difficult to assess; clinicians should assume conscious level and amnesia relate to injury and have a lower threshold for referral and neuro-imaging, since blunt head injuries in intoxicated patients are associated with a higher mortality.[35] Alcohol can also lead to headaches, nausea and irritability, making examination challenging. Five of the identified clinical decision support tools included 'drug/alcohol use' (HIDATq, HAYDEL, ATABAKI, CIDSS2, Brain Injury Guidelines). None of the most validated tools, PECARN, CHALICE or CATCH, include this as a clinical predictor.

### Age

All the identified clinical decision support tools in this review specify an age in which the tool can be used, for example, under 18 years, and this varies from age under 2 years to under 21 years. However, only two tools exclude younger age groups as a clinical predictor (HIDATq excludes those under 1 years and ATABAKI excludes those under 2 years). HAYDEL is only for children 5–17 years old and PECARN has a separate tool for those under 2 years, with different questions, more specific to that age group.[6] BUCHANICH, PredAHT and Pedi-BIRN are designed specifically for children under 3 years, and GREENES is for those under 2 years only; however, despite these tools aiming at the same age group, their clinical predictors still vary considerably. The authors of GREENES found that infants are at greater risk of intra-cranial hypertension due to their open fontanels and that those under 1 year with intracranial injuries are frequently asymptomatic, making management inherently difficult for clinicians.[22] It may be for this reason (as well as the increased likelihood of non-accidental injury) that children under 3 months were significantly more likely to receive a CT scan using the PECARN tool.

## DISCUSSION

This systematic mapping review identified and examined clinical criteria within existing hospital-based clinical decision support tools for children with minor head injury to determine their utility in paramedic practice, to inform the need for hospital conveyance. There are additional factors which require consideration in operationalising a tool for prehospital use, including the purpose of existing clinical decision rules and additional relevant clinical factors.

### Challenges of selecting a tool for use in prehospital care

None of the existing hospital-based clinical decision support tools can be translated directly into paramedic practice. No single hospital-based clinical decision support tool included all 18 clinical criteria, and there are potentially additional criteria that are not included in any of the tools that may be useful specifically in prehospital care; however, this needs to be explored in subsequent work. Most hospital-based clinical decision support

tools are designed to identify children who need a CT scan, although this is not actually the clinical endpoint of interest. The reason for doing a CT scan is to seek intracranial injury that requires neurosurgery. Arguably, if a child receives no active treatment, then a CT scan is unnecessary, even if it does show intracranial injury[36] especially as paediatric CT scanning results in a significantly increased lifetime risk of radiation-induced fatal malignancy.[37] The aim of a prehospital clinical decision support tool designed for use by paramedics (following additional research) will likely be to identify head-injured children who can safely be left at scene, and *do not* need conveyance. Therefore, the tool will 'rule out' serious head injury and be highly sensitive, since if all answers are negative in a highly sensitive tool, the tool can be used to rule out clinically significant brain injury. This is accepting that there are complicating issues specific to prehospital care, such as local ambulance policy, time of day, time since injury, complex wound closure, safeguarding, parental concern and parental capability, that need to be considered and may warrant conveyance, despite there being minimal concern of a clinically significant head injury.[38]

All the existing hospital-based tools address diverse questions, age groups, injury severities, and outcomes and have been derived, evaluated, and validated in a different way. In comparison with CHALICE, CATCH and other hospital-based tools, PECARN identifies children with a very low risk of brain injury who *do not* need a scan, which is an important concept, because it considers who a clinician *does not* need to scan, rather than who they *do*.[6] PECARN was derived using two large multicentre cohorts with over 40 000 children and is extremely sensitive, particularly for infants. A tool designed for use by paramedics in the prehospital setting would likely aim to follow this approach and identify children who can safely be left at scene and *do not* need conveyance, due to sustaining a trivial or minor head injury that is very low risk of clinically significant brain injury, similarly to Kuppermann *et al*'s PECARN,[6] though this needs further consideration and study. The role of ambulance paramedics includes the ability to make autonomous decisions on the safe discharge of patients at scene to prevent unnecessary conveyance to the ED. This is similar to in-hospital clinicians whose role it is to discharge patients from the ED when it is safe to do so. Therefore, the in-hospital clinical decision support tools for the assessment and management of children with head injury will share many similarities, particularly if the tools are being used in triage. PECARN is also the only tool that considers the age of the child from the beginning and thereafter asks different questions relevant to the child's age.[6] Paramedics have reported difficulty using the NICE head injury guidance[4] for this reason, as they could not ask younger children about amnesia, headache and alcohol use.[38]

## Paramedic use of clinical decision support tools in current practice

Previous research demonstrates that paramedics find clinical decision tools useful in identifying alternative patient pathways for other patient groups, such as older adults following a fall.[39] Clinical decision tools are not a new concept and have been at the forefront of digital health solutions for more than 12 years.[40] In prehospital care, they are used by paramedics to manage risk and support referral to alternative care pathways and community-based care.[41] Additionally, the use of clinical decision tools in the prehospital setting by paramedics has been shown to significantly reduce the opportunity for human error, improve diagnostic accuracy, improve optimisation of resources, improve compliance with guidelines and improve overall quality and safety of care for patients.[42 43] One systematic review of randomised controlled trials (RCTs) compared clinical decision tools with usual care in the prehospital setting and concluded that there are too few RCTs to draw any firm conclusions in prehospital care, although there is a possibility that clinical decision tools increase diagnostic accuracy.[44] A more recent cluster RCT investigating the effectiveness of a clinical decision tool to increase the number of elderly fallers referred to a community falls team rather than ED found evidence that paramedics use clinical decision tools to justify current practice rather than to inform decisions. The study found that paramedics decided to leave patients at home first, and then used the clinical decision tool to justify their decision. The tool was found to be cost-effective and successful at changing paramedic practice.[45]

## What clinical criteria should be included

All 18 clinical criteria identified in this systematic mapping review are important to consider for a prehospital tool designed to support paramedics in their assessment and management of children with head injury; however, some may be difficult to operationalise in the prehospital setting. In addition, the hospital-based tools have varying cut-off points for different clinical criteria and which descriptor a prehospital tool should use requires further consideration. It is important to acknowledge that all clinical decision tools should be used alongside clinical judgement. While this review is intended to support the process of developing a clinical tool to support paramedic decision-making at scene, it is only an initial step, and it is recognised that considerably more research is required. It would only be reasonable to adapt in-hospital tools for prehospital use if it could be demonstrated that the tool was applied with similar reliability and accuracy to that which is currently achieved in hospital practice.

## Certainty of use

It is clear from the evidence that some clinical predictors are significant for identifying a potentially clinically important traumatic brain injury following a head injury and should be included in a prehospital tool, such as a post-traumatic seizure.[30] Signs of a skull fracture were

most likely to trigger a CT response over any other clinical criteria, since the risk of intracranial injury is significantly increased in the presence of a skull fracture. This is reflected in the NICE guidance for the management of head injury, which suggests a CT scan should be performed within an hour if there are signs of a skull fracture.[4] The tools indicated that specifically a 'large', or 'non-frontal' (particularly parietal)[22] or 'boggy' haematoma, especially in infants, is more indicative of significant intracranial injury, which is apparent in other research.[44] Infants under 6 months old have a higher likelihood of significant intracranial haemorrhage with haematomas.[46] Greenes and Schutzman[22] suggest that among asymptomatic head-injured infants, the risk of skull fracture and associated intracranial haemorrhage is correlated with both scalp haematoma size and location. In most ambulance trusts across the UK, local policy stipulates that any children under 2 years old should be conveyed to the ED, no matter how mild the presenting injury.[38] Abusive head trauma occurs more frequently in those under 1 year[47]; paramedics are in a unique position to identify suspected non-accidental injury since they attend the patient's home environment often in a time of crisis and have a professional duty to protect vulnerable persons.[48] This evidence supports the recommendation of including 'infants' as a clinical predictor for a prehospital tool designed to aid decision-making regarding conveyance, or to exclude those under 1 year entirely. Non-accidental injury is only a clinical predictor in two in-hospital clinical decision tools, however; perhaps this is because it is seen as a 'no-brainer' and children were often excluded from the study cohort, as they would always need a scan. A child with suspected non-accidental injury should always be conveyed to the ED by paramedics. Abuse tools such as those derived by Pfeiffer et al[15] and Cowley et al[49] include certain clinical predictors that can help paramedics in recognising non-accidental injury.

### Clinical predictors that may need intervention

The prehospital environment is very different to in-hospital and there are some clinical predictors that may warrant conveyance as they may need interventions that cannot be done prehospitably, such as observation and complex wound closure. In addition, there are factors that are outside of a paramedic's scope of practice that present too much risk to discharge at scene, and therefore warrant conveyance.

The number of vomits was variable across the tools; research suggests that children who vomit after a head injury is less significant than in adults and they do not necessarily have a serious brain injury.[50] Other literature supports this, suggesting that clinically important traumatic brain injuries are uncommon in children presenting with head injury with vomiting and a management approach of observation without immediate CT may be appropriate.[51] However, a period of observation is not always practical or possible for paramedics in the prehospital setting. This was reported by paramedics in

a survey exploring reasons why paramedics convey children with minor head injury to the ED.[38] However, most children with head injury (90%) who are admitted for observation are discharged without any further treatment being given.[52] Notably, *persistent* vomiting after head injury is indicative of a subdural haematoma or skull fracture; Harper et al[53] support this, concluding that discreet vomiting over four times was a significant risk factor for intracranial injury. Additionally, vomiting, combined with other clinical criteria such as headache and altered mental status, can indicate a more serious head injury.[51] This theme is clear with other clinical predictors such as LOC; children with minor blunt head trauma presenting to the ED with *isolated* LOC are at very low risk of clinically significant brain injury and do not routinely require CT scanning[54]; however, if the LOC was associated with other clinical predictors, the risk is increased. Any clinical predictor present in isolation is less likely to indicate clinically significant traumatic brain injury than the presence of multiple risk criteria.[51] This is similar for mechanism of injury; evidence implies that mechanism of injury alone is less predictive of an abnormal brain CT, compared with other clinical predictors.[55] Despite this, most paramedics would convey a head-injured child based solely on mechanism, despite them appearing well with no clinical concern.[38]

Fowler et al[56] found that attending to paediatric patients evokes anxiety and discomfort among paramedics, which often led to reluctance to initiate treatment and poorer care. This includes unnecessary transport to the ED. Approximately half of paramedic respondents in a survey reported feeling 'not at all confident' with completing a neurological assessment on a child following a head injury,[38] despite a focal neurological deficit being a strong indicator of intracranial injury.[18] GCS is inherently difficult to assess in a younger child; however, a reduced GCS is a high-risk factor for clinically significant intracranial injury.[26]

### Clinical predictors that infrequently result in intracranial injury

There is limited evidence to suggest vertigo as a standalone clinical predictor should be included in a decision tool for conveyance of children with head injury, and this needs further investigation.[27] Additionally, evidence is still conflicting regarding the risk of intracranial haemorrhage in children with a clotting impairment/anticoagulation,[57] despite this being a research priority since 2011.[58] Bressan et al[57] found that although children with a bleeding disorder who have sustained a head injury more often received a CT scan, their risk of intracranial haemorrhage was low. Their findings are echoed in other literature[59]; CTs are obtained twice as often in children with bleeding disorders, although intracranial haemorrhage occurred in only 1.1% of these patients, and they had symptoms that would have prompted the need for a CT anyway. New guidance from NICE[4] advocates a CT head scan within 8 hours for children on anticoagulant

treatment or within an hour if they have a bleeding/clotting disorder, independently of other indications. However, only 1 out of 31 children on anticoagulants in Bressan *et al*'s[57] and Lee *et al*'s[59] study combined was diagnosed with an intracranial haemorrhage, and that patient very obviously needed a CT from the outset. These clinical criteria need further consideration before being included in a prehospital tool for paramedic use.

### Research trajectory for a clinical decision tool designed to support paramedics in the assessment and management of children with head injury

The proposed development of this clinical decision support tool will follow the Medical Research Council guidance for complex interventions. The 'development' phase of the 'development–feasibility–evaluation–implementation' process should begin with an investigation and identification of the evidence base through a systematic review. The next stage will involve developing theory using interviews and a Delphi method with key stakeholders, with the aim of creating the new tool. Following this, the process and outcomes will be modelled before proceeding to a full-scale evaluation by assessing the acceptability and usability of the new tool by paramedics using simulated scenarios. Further evaluation will involve assessing the validity and test characteristics of the final tool to examine safety, without changing clinical practice. If validity and safety are confirmed, then a feasibility study for a subsequent RCT will follow to determine whether the introduction of the new support tool reduces conveyance to the ED for children with head injury, while maintaining safety and avoiding adverse clinical outcomes. It is acknowledged that even if the tool had very high sensitivity, it would need to be compared with clinical judgement.

### Limitations

The role of mapping reviews is to provide a descriptive account of the published literature; therefore, this should be considered when assessing the findings of the overall evidence synthesis. A direct comparison of the clinical decision tools is not possible because they address different questions, age groups, injury severities and outcomes. It is recognised that none of the evidence is from the prehospital environment, since there are no existing out-of-hospital clinical decision support tools designed to support the assessment and management of children with head injury, hence the need for this research. Therefore, the in-hospital publication of tools considering childhood injury is the only research available to draw conclusions from at this point. Elements of in-hospital research can be attributed to prehospital care since there are similarities; however, it is acknowledged that further research is required with prehospital staff in the prehospital environment following this work. Although rigorous methods of identifying all in-hospital tools have been followed, it is possible that some studies have been overlooked. This is particularly the case for international studies since this review included English-language articles only.

## CONCLUSIONS AND FUTURE RESEARCH

Clinical predictors that markedly increase the likelihood of neurosurgical intervention and should be considered for inclusion in a clinical decision support tool for use by paramedics are: signs of a skull fracture; a large, boggy or non-frontal haematoma (particularly in an infant); persistent discreet episodes of vomiting; a focal neurological deficit; GCS less than 15; a prolonged or worsening headache; a prolonged LOC; a post-traumatic seizure; amnesia in older children; suspicion of non-accidental injury; drug or alcohol use; and being under 1 year old. Clinical criteria that require further investigation include mechanism of injury, clotting impairment/anticoagulated, vertigo, length of time of LOC, number of vomits and description of what constitutes a neurological deficit in younger children. Any clinical predictor present in isolation is unlikely to indicate clinically significant traumatic brain injury.

There are likely to be additional clinical criteria that are relevant to paramedic assessment and practice, which are not included in any of these tools. Such criteria would need to be identified and explored in future qualitative work involving interviewing paramedics about what influences them when deciding to convey children with minor head injury to the hospital. None of the existing hospital-based clinical decision support tools can be directly implemented into paramedic practice; however, elements from each of the tools can be adapted to create a new tool specific for paramedics in prehospital care. Future qualitative research is needed to investigate paramedics' thoughts on a clinical decision tool designed to support them in their assessment and management of head-injured children. Additionally, future research using Delphi methods is required to build a consensus among study matter experts and users to enable the development of a new clinical decision tool to support paramedics in safely assessing and managing children with minor head injury.

**Author affiliations**
[1]University of the West of England, Bristol, UK
[2]Faculty of Health and Applied Sciences, University of the West of England Bristol, Bristol, UK
[3]Health and Life Sciences, University of the West of England, Bristol, UK
[4]Academic Department of Emergency Care, The University Hospitals NHS Foundation Trust, Bristol, UK
[5]Faculty of Health & Life Sciences, University of the West of England, Bristol, UK

**Contributors** JRB, SV and ML supervised the work. AP wrote the protocol, conducted the review and drafted the manuscript with input from JRB, SV, ML, JB and JS. JB double-screened the abstracts and full texts. PS helped to design the search terms and conduct the search. All authors interpreted and analysed the results. All authors discussed the results, contributed to and approved the final manuscript. AP is the guarantor.

**Funding** This work is supported by Health Education England/National Institute for Health Research Integrated Clinical and Practitioner (ICA) Programme's Doctoral Clinical and Practitioner Academic Fellowship (DCAF) Scheme (award number 25215).

**Disclaimer** The views expressed are those of the authors and do not necessarily represent those of Health Education England, the National Institute for Health Research or Department of Health.

**Competing interests** None declared.

**Patient and public involvement** Patients and/or the public were involved in the design, or conduct, or reporting, or dissemination plans of this research. Refer to the Methods section for further details.

**Patient consent for publication** Not required.

**Ethics approval** Not applicable.

**Provenance and peer review** Not commissioned; externally peer reviewed.

**Data availability statement** Data sharing not applicable as no datasets generated and/or analysed for this study.

**Author note** Transparency statement: This manuscript is an honest, accurate and transparent account of the study being reported. No important aspects of the study have been omitted and any discrepancies from the study as originally planned have been explained.

**ORCID iDs**
Alyesha Proctor http://orcid.org/0000-0001-5763-5419
Sarah Voss http://orcid.org/0000-0001-5044-5145

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
