## [Reviewer comments · BMJ Open]

ARTICLE DETAILS

TITLE (PROVISIONAL)	Which elements of hospital-based clinical decision support tools for the assessment and management of children with head injury can be adapted for use by paramedics in pre-hospital care? A systematic mapping review and narrative synthesis.
AUTHORS	Proctor, Alyesha; Lyttle, Mark; Billing, Jedd; Shaw, Pauline; Simpson, Julian; Voss, Sarah; Bengner, Jonathan

VERSION 1 – REVIEW

REVIEWER	Hoffman, Jerome University of California Los Angeles David Geffen School of Medicine, EM
REVIEW RETURNED	14-Aug-2023

GENERAL COMMENTS	While I have concerns about treating all the cited research as carrying the same weight, I don't believe this is a major problem. Your search methods are fine, and the search comprehensive. In addition, you appropriately note that none of the cited studies address your research question (b/c none of those studies were done in a pre-hospital setting), and that firm conclusions must be limited. But I wonder how you can come to any conclusions without first attempting to define what should be the role of pre-hospital providers, and how that is different from in-hospital physicians. Do you believe they should be performing the same function (deciding which patient needs a CT scan, per your manuscript)? If not, how do these in-hospital publications address your research question? Even more important, to my mind, is the question of to what extent any of the decision instruments add to or improve clinical judgment. And it is hard to imagine that any clinician (including pre-hospital providers) would leave at the scene any patient with any of the clinical characteristics you mention in your conclusions (eg an abnormal GCS, a neurologic deficit, suspicion of abuse, etc). And if clinical judgment is important, shouldn't you be addressing what is the appropriate role of judgment by pre-hospital providers, vs by in-hospital specialists? Finally, it seems to me the potential value of decision instruments is to allow providers (albeit in-hospital providers, in the studies cited) to feel comfortable deciding that certain groups of patients DO NOT require any further evaluation ... rather than telling them that a patient who is obviously at risk does need more. Because head trauma can lead to catastrophic outcomes, clinicians have historically done an extensive (and potentially harmful) work-up in many patients who were clearly at very little risk; isn't this where
--

	DCIs have potential utility (more than in saying that a child at risk of abuse deserves to be taken to hospital)?
--	---

REVIEWER	Delbourgo Patton, Caroline Albert Einstein College of Medicine, D. Samuel Gottesman Library
REVIEW RETURNED	18-Aug-2023

GENERAL COMMENTS	This is a well-done mapping review. The literature review methods are thorough and thoughtful, including a good selection of databases and appropriate incorporation of grey literature and hand searching of article reference lists. It may be useful to specify the grey literature sources that were searched in the manuscript itself in the manuscript itself since the protocol specifically mentions Open Grey in addition to soliciting input on appropriate grey literature.
--

REVIEWER	Monteiro, Luis CINTESIS
REVIEW RETURNED	21-Aug-2023

GENERAL COMMENTS	Dear authors, Congratulations: the manuscript addresses a very important topic with clinical implications. Just minor suggestions: --can you enlarge the words and numbers of the PRISMA diagram? -- Can you please elaborate on future research? Be specific: methods, scope.
--

VERSION 1 – AUTHOR RESPONSE

Provide clarity on what should be the role of pre-hospital clinicians and how this differs from in-hospital clinicians. Is it deciding that a CT scan is needed. Provide clarity on how the in-hospital publications are relevant to the pre-hospital setting.	On pages 20 and 21 we consider whether the aim should be to identify children who do not need conveyance, due to sustaining a trivial or minor head injury that has a very low risk of clinically significant brain injury, as opposed to identifying those who do need conveyance for a potential CT scan or observation or both. We have included further detail on the role of paramedics and provided clarity on why hospital publications are relevant to the pre-hospital setting on pages 20 and 23.
Provide clarity on what extent the clinical decision makers are important in deciding whether to take a child to hospital. Impact of clinical judgement.	Sentence added on page 21 to acknowledge that all clinical decision tools should be used alongside clinical judgment.
Is the aim of the tool to decide whether certain children DO NOT need to go to hospital, rather than those that may need further work up.	The aim of the tool is to support paramedics to safely assess and manage children at scene, by identifying those who have sustained a minor/trivial head injury that does not require further intervention. However, it also has the potential to identify children who may need conveyance for observation, complex wound closure, a safeguarding concern, and those who may need a CT scan. The tool could have several potential outcomes, for example attend an ED, attend a MIU, discharge at scene with written worsening advice, etc. This needs to be explored in further work.
Enlarge the words on the PRISMA diagram	Increased font size from 9 to 11.
Elaborate on future research being specific about methods	We have added text within the conclusion and future research subheading to describe planned future research, including the intended methods.

VERSION 2 – REVIEW

REVIEWER	Hoffman, Jerome University of California Los Angeles David Geffen School of Medicine, EM
REVIEW RETURNED	03-Nov-2023

GENERAL COMMENTS	I appreciate the effort that went into this study, as well as the appropriate methodology used for the literature review. I remain concerned about three major issues, however. The first of these is perhaps the least important, although I don't believe it is minor. You acknowledge that none of the cited studies were done with paramedics; thus adapting them for use by paramedics would only be reasonable if there was good reason to believe that paramedics could apply them with the same reliability
---

	and accuracy as in-hospital doctors. I don't believe there is any such evidence, and in fact you yourselves note at least some ways in which this is surely not the case. Second, you seem to be very ambiguous about whether you are proposing that a clinical decision instrument (CDI) can be used to predict who can safely be left in the field, as opposed to who must be transported, as opposed to both. You conflicting assertions about this at various times. This is an important issue, as a CDI that is very specific (the first type of CDI) is almost certainly going to be insensitive, while one with high sensitivity (needed for the second type of CDI) will surely be non-specific. Perhaps most important, adding a sentence about "combined with clinical judgment" does nothing to address my concern that you provide no evidence that using any proposed CDI – either alone or "in combination with" judgment would improve decision-making. In order to address this, I believe, you would need to study, or at the very least hypothesize about, how paramedics currently perform when deciding whether or not to transport ... and then study (or at the very least hypothesize about) how adding a CDI would be expected to improve the quality of their decisions.
--	---

VERSION 2 – AUTHOR RESPONSE

Comments	Changes made
I appreciate the effort that went into this study, as well as the appropriate methodology used for the literature review. I remain concerned about three major issues, however. The first of these is perhaps the least important, although I don't believe it is minor. You acknowledge that none of the cited studies were done with paramedics; thus, adapting them for use by paramedics would only be reasonable if there was good reason to believe that paramedics could apply them with the same reliability and accuracy as in-hospital doctors. I don't believe there is any such evidence, and in fact you yourselves note at least some ways in which this is surely not the case.	Thank you for your positive feedback. We have addressed this in the discussion section under the subheading 'What clinical criteria should be included' as follows: Whilst this review is intended to support the process of developing a clinical tool to support paramedic decision making at scene, it is only an initial step, and it is recognised that considerably more research is required. It would only be reasonable to adapt in-hospital tools for pre-hospital use if it could be demonstrated that the tool was applied with similar reliability and accuracy to that which is currently achieved in hospital practice.
Second, you seem to be very ambiguous about whether you are proposing that a clinical decision instrument (CDI) can be used to predict who can safely be left in the field, as opposed to who must be transported, as opposed to both. Your conflicting assertions about this at various times. This is an important issue, as a CDI that is very specific (the first type of CDI) is almost certainly going to be insensitive, while	Thank you for your comment. We have now clarified in the review that the ultimate aim of the clinical decision tool (following further research) will likely be to identify children who can safely be left at scene ("rule-out" serious head injury). This is with the proviso that there are certain rule-in criteria that are specific to the pre-hospital setting and may still require hospital conveyance, such as wound closure, safeguarding or parental

one with high sensitivity (needed for the second type of CDI) will surely be non-specific.	concerns. It is recognised that this needs further research, which we have planned. This is mainly within the discussion section under the subheading 'Challenges of selecting a tool for use in pre-hospital care' but has been updated throughout.
Perhaps most important, adding a sentence about "combined with clinical judgment" does nothing to address my concern that you provide no evidence that using any proposed CDI – either alone or "in combination with" judgment would improve decision-making. In order to address this, I believe, you would need to study, or at the very least hypothesize about, how paramedics currently perform when deciding whether or not to transport ... and then study (or at the very least hypothesize about) how adding a CDI would be expected to improve the quality of their decisions.	We have added text within the discussion section under a new subheading 'paramedic use of clinical decision support tools in current practice' to describe how a clinical decision support tool could act to support and improve paramedic decision making. There is limited pre-hospital research evidence about how adding a clinical decision tool improves quality of paramedic decisions, however we are planning to undertake further work to address this next.
Please ensure you attach the search strategy as a supplementary material file.	This has been resubmitted as a supplementary material file as requested.

VERSION 3 – REVIEW

REVIEWER	Hoffman, Jerome University of California Los Angeles David Geffen School of Medicine, EM
REVIEW RETURNED	16-Dec-2023

GENERAL COMMENTS	I greatly appreciate the excellent revisions you have made; the manuscript is now greatly improved. This, along with your careful methodology, makes a strong argument for publication. The only further suggestion I would make, regarding the paper itself, is that you consider including in your discussion a comment about what next steps would be necessary in the process of creating a CDI – including first a derivation set (based on your work) that when tested would then still only be able to generate a hypothesis, and then a second (separate) study to validate whatever CDI has been derived. In addition, you would want to note that even if such a CDI appeared to have very high sensitivity, it would need to be compared to clinical judgment. This latter comment is the basis for my ongoing concern about the importance of what you have found – it is hard to imagine, for example, that any paramedics would leave at the scene a child who has any single one of your "clinical criteria [which] should be included." (I do acknowledge that this entire process could conceivably end up validating an equally sensitive CDI that has added value because it also has higher specificity than judgment alone ... although I suspect this to be quite unlikely.) But I will leave it to editors to decide about this.
---

	[I have checked "accept" as my recommendation, because my choices are limited. But I do believe it is the editors of the J, and not I, who have to decide on the importance of publishing this paper.
--	--

VERSION 3 – AUTHOR RESPONSE

Include a comment in the discussion about what next steps would be necessary in the process of creating a CDST.	Thank you for your comment and recommending the article to be accepted for publication following the revisions that have been made. We have included text within the discussion section regarding planned next steps for this CDST. We have also acknowledged that the tool would need to be compared to clinical judgement.
--	---